**Data Availability Statement:** All relevant data are within the manuscript and its Supporting information files.

# Preparing for the interviewing process during Coronavirus disease-19 pandemic: Virtual interviewing experiences of applicants and interviewers, a systematic review

**Sonal Chandratre** [1,2]*, **Aamod Soman** [3]

**1** Medical College of Wisconsin-Central Wisconsin Regional Campus, Wausau, Wisconsin, United States of America, **2** Department of Pediatrics, Ascension Saint Michael's Hospital, Stevens Point, Wisconsin, United States of America, **3** Department of Internal Medicine, Ascension Saint Michael's Hospital, Stevens Point, Wisconsin, United States of America

* sonal.chandratre@ascension.org

## Abstract

### Purpose

Coronavirus disease-19 (COVID-19) has forced upon all academic institutions to conduct virtual interviewing (VI) instead of face-to-face interviewing (FTFI) this interviewing cycle. The purpose of this systematic review was to understand the process of VI, its effectiveness as an alternative to FTFI, and the experiences of applicants and institutions with VI. We also share best practice strategies for applicants and institutions in VI preparation.

### Method

PubMed/MEDLINE, Cochrane Library of Systematic Reviews, Web of Science Core Collection, Scopus and CINAHL databases were searched through May 2020. Articles in English evaluating the effectiveness of VI were included, without applying any date limits. Two reviewers selected articles and extracted data.

### Results

Of the 934 articles screened, 22 articles underwent full-text article analysis to include 15 studies. There were 4 studies that reported the use of VI as a screening tool. 11 studies completely replaced FTFI with VI. Most applicants could appropriately convey themselves through VI. Most applicants and interviewing programs expressed reservations about VI's use as an alternative to FTFI.

### Conclusion

There is dearth of evidence supporting the efficacy of VI. There is an opportunity for potential research at multi-institutional level to gain better understanding of the efficacy of VI. The knowledge obtained from this systematic review has the potential of helping applicants and

**Funding:** The authors received no specific funding for this work.

**Competing interests:** The authors have declared that no competing interests exist.

institutions in preparing for VI process. Additionally, authors propose supportive strategies to help prepare applicants and institutions for VI.

## Introduction

These are unprecedented times. Coronavirus disease-19 (COVID-19) has had an immense impact on the world. Medical education is also affected by COVID-19 with multiple enforced changes in medical education potentially increasing anxiety and stress in medical students [1]. Due to the evolving pandemic, there are travel restrictions hindering the 2021 interviewing cycle. Interview process is crucial for securing a position. With the uncertainty surrounding COVID-19, several institutions are compelled to transition from face-to-face interviewing (FTFI) to virtual interviewing (VI) for the upcoming interviewing season. Although VI maybe explored in nonmedical disciplines, there are limited reports in the existing medical literature describing the utilization of VI in interviewing [2–18]. Traditionally, FTFI remains to be a preference for candidate selection process. This preference may be attributed to the need for assessing interpersonal communication skills of applicants and FTFI may be perceived as an effective method over VI. This review article focuses on synchronous interviewing wherein there was a live interview between the interviewee and the interviewer. However, the Association of American Medical Colleges has previously evaluated the use of a Standardized Video Interview (SVI) delivered asynchronously wherein the interviewee submitted prerecorded interview answering a set of questions uniform to all interviewees designed to assess applicants' interpersonal and communication skills and knowledge of professionalism as an innovative tool for applicant screening by institutions for their FTFI [19]. Applicants had generally negative reactions to the SVI [20] and approximately half of the responding programs who utilized SVI in their selection process reported its use mainly in tiebreaker situations between applicants with similar profiles [21]. There are many potential advantages of VI. Applicants and hosting institutions, both benefit economically by cutting down on interviewing costs and additionally save valuable time otherwise spent in the interviewing process disrupting patient care and other commitments.

VI can be challenging for institutions and applicants, given its limited use. Although some institutions have previously evaluated the value of VI voluntarily, COVID-19 has forced all institutions to implement VI regardless of previous experience conducting VI.

Both applicants and institutions apprehensively search best practice strategies for VI. Our systematic review seeks to address this concern by evaluating studies from literature to understand the process of VI, its effectiveness as an alternative to FTFI, and the experiences of applicants and institutions with VI. The knowledge attained through this systematic review and the evidence-based strategies provided by authors will support applicants and institutions in developing best practices for an efficient future VI process in their own institutions.

## Methods

We conducted a systematic review of the peer-reviewed literature to explore the use of VI in the interviewing process. We sought to understand the process of VI used, whether it was an effective alternative to FTFI and the experiences of applicants and institutions with VI. This review followed the Preferred Reporting Items for Systematic Reviews and Meta-Analyses statement [22].

## Search strategy

In May 2020, S.C and A.S conducted the database search and applied a systematic search method to 6 major databases: PubMed/MEDLINE, Cochrane Library of Systematic Reviews, Web of Science Core Collection, Scopus, and CINAHL. We used Medical Subject Headings (MeSH) and combinations of search terms keywords including virtual, online, videoconferencing, residency and fellowship in combination with interviews. See S1 Appendix. MeSH Used For PubMed Database. We applied no publication date constraints. Additionally, we hand searched the reference lists of articles included in our full-text analysis. The search covered all articles published on or prior to May 31st, 2020 in English, German and French.

## Eligibility criteria

We included all articles published in English language, published in peer reviewed journals, and those who had data on synchronous web-based interviews for undergraduate school, residency and fellowship from all healthcare fields. We excluded the articles that did not meet these criteria. In addition to original articles, we also included those perspectives, viewpoints, commentaries, letters to the editor and narrative essays that included data on using VI for recruitment. We excluded systematic reviews, meta-analyses, conference abstracts, dissertations, errata, podcasts, video reviews, books and book reviews.

## Study selection

The database search was performed by S.C and A.S together. We identified and excluded duplicates. Two reviewers (S.C. and A.S.) reviewed the titles and abstracts of the retrieved articles. We then selected articles for full-text review. In event of an ambiguity about the inclusion of a study, authors discussed the study to take final decision on the study's eligibility with consensus.

## Data extraction

We manually extracted information from the included studies into an Excel sheet. Study synthesis included the following: author information, title of the study, year of publication, year the study was performed, country of origin, article type, study design, aim of the study, sample size, participant level of training, number of interviewers, platform used for the interview, reported technical issues, number of cycles VI used for recruiting, inclusion of a control group, provision of program information, provision of a hospital tour, provision of interaction with program trainees, interaction with peer interviewees, use of VI as a screening tool, presence of data on FTFI during the same interviewing season.

## Quality assessment

Two reviewers (S.C. and A.S.) independently assessed study quality using National Institute of Health's Quality Assessment Tool for Observational Cohort and Cross-Sectional Studies, a tool with 14 questions [23].

## Results

Our initial database search identified 1,103 articles (PubMed: 395, Cochrane: 128, Scopus: 302, Web Of Science: 180, CINAHL: 98). Seven studies were additionally selected from the reference list of included studies that underwent a full review. After removing 176 number of duplicates there were a total of 934 articles. Both authors screened the titles and abstracts of the remaining 934 number of articles to determine whether they met the eligibility criteria. Total

of 22 studies were finally included for full-text analysis. Authors accessed the included articles for further screening and discussed each article together. After performing full-text article analysis, 7 studies were excluded; three studies had data on asynchronous, pre-recorded applicant videos [19–21] as compared to the focus of this article of synchronous VI and 4 studies were devoid of data (2 letter to the editor [6, 8] and 2 perspectives [24, 25]). This ultimately resulted in a total of 15 studies for this review. See Fig 1.

## Quality assessment

There were 2 studies with poor rating [10, 18], 11 studies with fair rating [2–4, 9, 11–17] and 2 studies with good rating [5, 7] obtained from 2 reviewers who performed the rating individually. See Fig 2.

## Study characteristics

Study characteristics are represented in Fig 3. The oldest study was identified from the year 2000 [18] and the most recent from the year 2020 [2]. Of the total 15 included studies, most studies (n = 14) were descriptive (1 cohort [5] and 13 survey designs [2–4, 7, 9–14, 16–18]). One study was a randomized controlled crossover study [15]. There were 14 studies originating in United States [2–5, 7, 9–12, 14–18] and 1 from Australia [13]. Seven studies were published as original articles [3, 5, 7, 9, 11, 13, 15], 1 as a brief report [14], 4 as letters to the editor [2, 4, 16, 18], 2 as perspectives [10, 11] and 1 as a viewpoint [17]. Participants from 5 studies were residents applying for a fellowship (1 surgical oncology [2], 1 pediatric surgery [3], 1 aesthetic plastic surgery [4], 1 adult reconstruction [7], 1 gastroenterology [12]), 8 were medical students applying for residency (1 each for anesthesiology [9], urology [15], ophthalmology [16], plastic surgery [17], family medicine [14] and 2 for internal medicine [10, 18]), two for medical school [5, 13], and 1 for pharmacy school [11]. There was only one study that occurred during COVID-19 pandemic [2]. All other studies were performed prior to COVID-19. Most studies demonstrated the use of VI as a single occurrence in their institution [2–4, 9–12, 14–18], one study reported for 2 years [13] and two studies reported using VI for 3 consecutive years [5, 7].

## VI: Screening tool or a replacement to FTFI?

There were 4 studies that reported use of VI as a screening tool to assess applicants for FTFI [3, 4, 11, 14]. See Fig 4. Chandler et al., found that nearly half of their applicants for pediatric surgery fellowship (n = 20, 87% response rate) and all of their interviewing faculty (n = 3) reported VI to be a helpful screening tool however 80% applicants and all of their faculty disagreed or strongly disagreed that VI could substitute FTFI [3]. Miotto (n = 8, 57% response rate) reported that majority of their applicants applying for aesthetic plastics fellowship along with the faculty thought that VI was beneficial as a screening tool but did not report findings on what they felt about it replacing FTFI [4]. Temple reported that VI as a screening tool helped their pharmacy school decrease the number of FTFI and saved them a considerable amount of on-site interviewing time, however their experience was limited due to lack of data about applicants' opinion on VI [11]. Edje et. al.(n = 9, 90%), demonstrated that 78% of their applicants and 82% of the faculty were in favor of VI as a screening tool for the interviewing process but neither of them were comfortable using VI as a replacement for FTFI [14].

There were 11 studies that completely replaced FTFI by VI [2, 4, 5, 7, 9, 10, 12, 13, 15–18]. Vining et. al., (n = 16, 81% response rate), reported that majority of their surgical oncology fellowship applicants (68%) and faculty (50%) expressed a preference for FTFI [2]. Looking at the final acceptance rates, Ballejos et al. in a single institutional study (n = 96, 12.8% response rate)

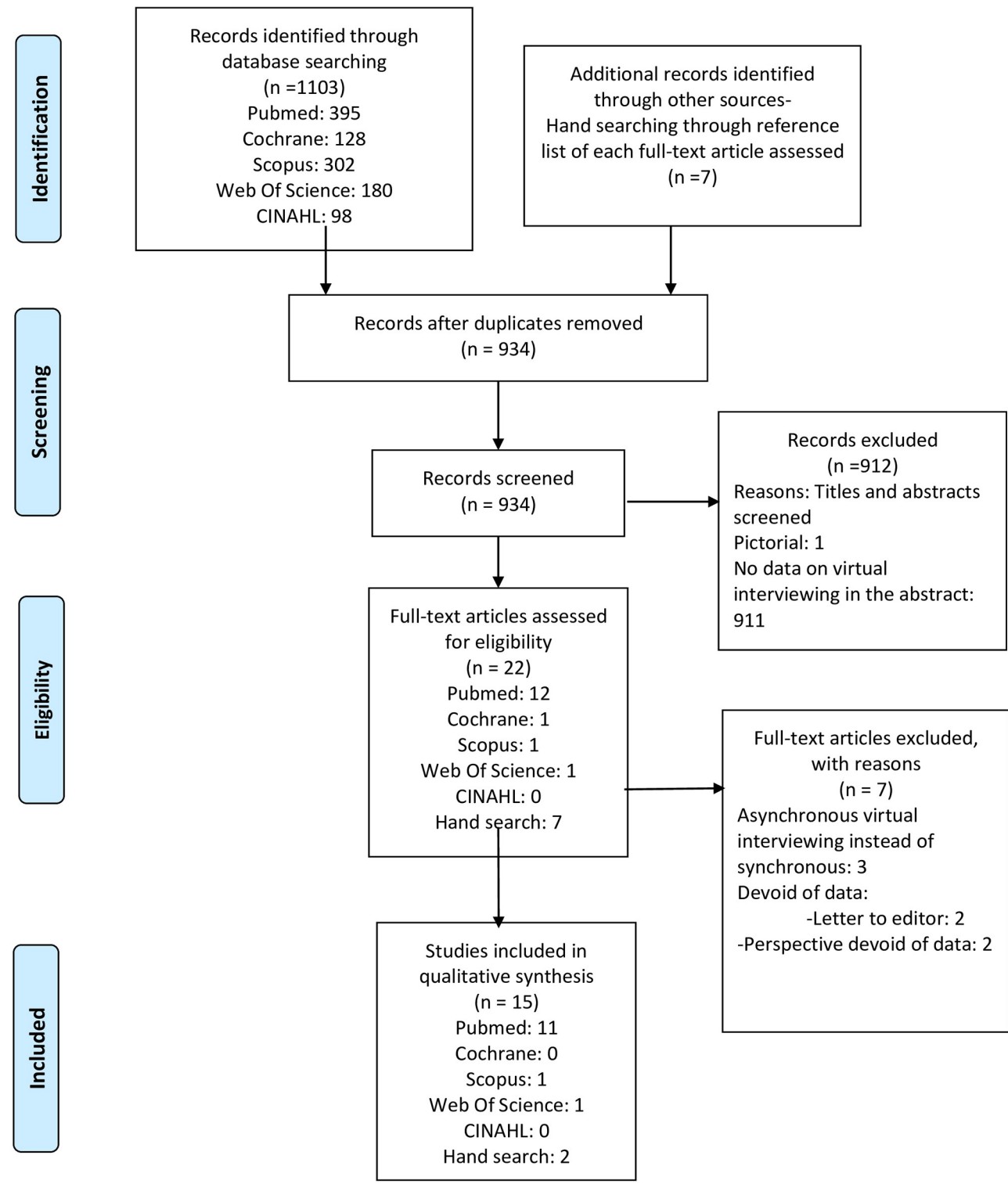

**Fig 1. Study selection and review process for systematic review in virtual interviewing.**

| First Author, Year study published, (Ref. no.) | Q1 | Q2 | Q3 | Q4 | Q5 | Q6 | Q7 | Q8 | Q9 | Q10 | Q11 | Q12 | Q13 | Q14 | Reviewer 1 | Reviewer 2 |
|---|---|---|---|---|---|---|---|---|---|---|---|---|---|---|---|---|
| Vining, 2020, (2) | Y | Y | Y | Y | N | N | Y | NA | Y | N | Y | N | NA | N | Fair | Fair |
| Chandler, 2018, (3) | Y | Y | Y | Y | N | N | Y | NA | Y | N | Y | N | NA | N | Fair | Fair |
| Miotto, 2018, (4) | Y | Y | Y | Y | N | N | Y | NA | Y | N | Y | N | NA | N | Fair | Fair |
| Ballejos, 2018, (5) | Y | Y | N | Y | N | Y | Y | NA | Y | Y | Y | N | CD | Y | Good | Good |
| Healey, 2017, (7) | Y | Y | Y | Y | N | N | Y | NA | Y | Y | Y | N | NA | N | Good | Good |
| Vadi. 2016, (9) | Y | Y | Y | Y | N | N | Y | N | Y | N | Y | N | NA | N | Fair | Fair |
| Williams, 2015, (10) | N | Y | N | Y | N | N | Y | NA | Y | N | N | N | NA | N | Poor | Poor |
| Temple, 2014, (11) | Y | Y | Y | Y | N | N | Y | NA | N | N | Y | N | NA | N | Fair | Fair |
| Daram, 2014, (12) | Y | Y | Y | Y | N | N | Y | NA | Y | N | Y | N | NA | N | Fair | Fair |
| Tiller, 2013, (13) | Y | Y | N | Y | N | N | Y | NA | N | Y | Y | N | NA | Y | Fair | Fair |
| Edje, 2013, (14) | Y | Y | N | Y | N | N | Y | NA | Y | N | Y | N | NA | N | Fair | Fair |
| Melendez, 2012, (17) | Y | Y | N | Y | N | N | Y | NA | Y | N | Y | N | NA | N | Fair | Fair |
| Pasadhika, 2012, (16) | Y | Y | N | Y | N | N | Y | NA | Y | N | Y | N | NA | N | Fair | Fair |
| Shah, 2011, (15) | Y | Y | Y | Y | N | N | Y | NA | Y | N | Y | N | NA | N | Fair | Fair |
| Liman, 2000, (18) | Y | Y | CD | Y | N | N | Y | NA | CD | N | N | N | NA | N | Poor | Poor |

**Fig 2. Quality ratings of included studies according to NIH quality assessment tool for observational studies.** Note—NIH = National Institutes of Health, NR = not reported, CD = cannot determine, NA = not applicable. Q1. Was the research question or objective in this paper clearly stated? Q2. Was the study population clearly specified and defined? Q3. Was the participation rate of eligible persons at least 50%? Q4. Were all the subjects selected or recruited from the same or similar populations (including the same time period)? Were inclusion and exclusion criteria for being in the study prespecified and applied uniformly to all participants? Q5. Was a sample size justification, power description, or variance and effect estimates provided? Q6. For the analyses in this paper, were the exposure(s) of interest measured prior to the outcome(s) being measured? Q7. Was the timeframe sufficient so that one could reasonably expect to see an association between exposure and outcome if it existed? Q8. For exposures that can vary in amount or level, did the study examine different levels of the exposure as related to the outcome (e.g., categories of exposure, or exposure measured as continuous variable)? Q9. Were the exposure measures (independent variables) clearly defined, valid, reliable, and implemented consistently across all study participants? Q10. Was the exposure(s) assessed more than once over time? Q11. Were the outcome measures (dependent variables) clearly defined, valid, reliable, and implemented consistently across all study participants? Q12. Were the outcome assessors blinded to the exposure status of participants? Q13. Was loss to follow-up after baseline 20% or less? Q14. Were key potential confounding variables measured and adjusted statistically for their impact on the relationship between exposure(s) and outcome(s)?

concluded that VI did not negatively impact the overall acceptance of applicants to a medical school program [5]. Healy et al., (n = 47, 90% response rate) reported that 89% applicants shared that VI met their expectations [7]. Vadi et al., also completely replaced FTFI with VI in selecting their candidates for an anesthesiology residency program (n = 42) and showed that VI did not have any negative influence on the final acceptance rates [9]. Williams et al., conducted VI for selecting medical students for internal medicine residency and all applicants (n = 6) reported a positive experience with VI although they did not match any applicants who were interviewed by VI (n = 8) [10]. Daram et al., reported that 81% of their applicants for their gastroenterology fellowship (n = 16) met or exceeded their expectations and 25% applicants reported that VI was equal to or better than FTFI [12]. Tiller et al., experimented with VI in

| First Author, Year study published, (Ref. no.) | Year(s) in which the study was performed | Applicant level of training | Institution/ Program's specialty setting | Study design | Country study was performed | Article category of publication | Applicant sample size (n) with (Response Rate%) | Interviewing faculty sample size (n) with (Response Rate%) | Number of years of implementati on |
|---|---|---|---|---|---|---|---|---|---|
| **During coronavirus disease-19** | | | | | | | | | |
| **Vining, 2020, (2)** | 2020 | Resident | Surgical oncology fellowship | Survey | United States of America | Letter to editor | 16 (80%) | 12 (92%) | 1 |
| **Before coronavirus disease-19** | | | | | | | | | |
| **Chandler, 2018, (3)** | 2017 | Resident | Pediatric surgery fellowship | Survey | United States of America | Original report | 20 (87%) | 3 (100%) | 1 |
| **Miotto, 2018, (4)** | NR | Resident | Aesthetics Plastics fellowship | Survey | United States of America | Letter to editor | 8 (57%) | NR | 1 |
| **Ballejos, 2018, (5)** | 2017 | Premedical | Medical school | Cohort | United States of America | Original report | 96 (12.8%) | NR | 3 |
| **Healey, 2017, (7)** | 2017 | Resident | Adult reconstructi on orthopedics fellowship | Survey | United States of America | Topics in training | 47 (90%) | 8 (NR) | 3 |
| **Vadi. 2016, (9)** | 2014, 2015 | Medical student | Anesthesiol ogy residency | Survey | United States of America | Original report | 42 (79%) | 6 (NR) | 1 |
| **Williams, 2015, (10)** | 2014 | Medical student | Internal Medicine residency | Survey | United States of America | Perspective | 8 (66.6%)) | 4 (NR) | 1 |
| **Temple, 2014, (11)** | 2013 | Pre-pharmacy | Pharmacy school | Survey | United States of America | Perspective | 24 (NR) | 9 (NR) | 1 |
| **Daram, 2014, (12)** | 2013 | Resident | Gastroenter ology fellowship | Survey | United States of America | Original report | 16 (66.6%)) | 1 (NR) | 1 |
| **Tiller, 2013, (13)** | 2011, 2009 | Premedical | Medical and Dental school | Survey | Australia | Original report | 119 (41%) | 78 (NR) | 2 |
| **Edje, 2013, (14)** | 2011, 2012 | Medical student | Family medicine residency | Survey | United States of America | Brief report | 9 (90%) | 6 (100%) | 1 |
| **Melendez, 2012, (17)** | 2009 | Medical student | Plastic surgery residency | Survey | United States of America | Viewpoint | 10 (NR) | 1 (NR) | 1 |
| **Pasadhika, 2012, (16)** | 2010 | Medical student | Ophthalmol ogy residency | Survey | United States of America | Letter to editor | 18 (77%) | 16 (94%) | 1 |
| **Shah, 2011, (15)** | 2010 | Medical student | Urology residency | Randomized control trial with cross over | United States of America | Original report | 33 (95%) | 6 (NR) | 1 |
| **Liman, 2000, (18)** | 1999 | Medical student | Internal medicine residency | Survey | United States of America | Letter to editor | 8 (NR) | 1 (NR) | 1 |

**Fig 3. Study characteristics of virtual interviewing.** Note—NR = not reported.

| First Author, Year study published, (Ref. no.) | VI used as a screening tool | VI used as a substitute to FTFI | Control group | Digital platform used | Applicants could convey themselves well through VI | Study conclusion on "Can VI replace FTFI?" |
|---|---|---|---|---|---|---|
| **During coronavirus disease-19** | | | | | | |
| **Vining, 2020, (2)** | No | Yes | No | Zoom | Yes | Yes |
| **Before coronavirus disease-19** | | | | | | |
| **Chandler, 2018, (3)** | Yes | No | No | Skype | Yes | No |
| **Miotto, 2018, (4)** | Yes | No | No | Vidyo | Yes | No |
| **Ballejos, 2018, (5)** | No | Yes | Yes | Skype | NR | No |
| **Healey, 2017, (7)** | No | Yes | No | Skype | Yes | No |
| **Vadi. 2016, (9)** | No | Yes | Yes | Skype/ FaceTime | NR | No |
| **Williams, 2015, (10)** | No | Yes | No | Skype | Yes | No |
| **Temple, 2014, (11)** | Yes | No | No | Survey/ FaceTime | NR | No |
| **Daram, 2014, (12)** | No | Yes | No | Skype | NR | Yes |
| **Tiller, 2013, (13)** | No | Yes | No | Skype | NR | No |
| **Edje, 2013, (14)** | Yes | No | Yes | Survey | NR | No |
| **Melendez, 2012, (17)** | No | Yes | Yes | Skype | NR | No |
| **Pasadhika, 2012, (16)** | No | Yes | No | Skype | NR | Yes |
| **Shah, 2011, (15)** | No | Yes | Yes | Skype | NR | No |
| **Liman, 2000, (18)** | No | Yes | No | Skype | NR | No |

**Fig 4. Highlights of studies that evaluated virtual interviewing.** Note—NR = not reported, VI = Virtual interviewing, FTFI = Face-to-face interviewing.

international applicants, (n = 119, response rate 41%), and reported that 76% (n = 89) of them agreed that VI was effective [13]. Shah et al., evaluated the use of VI in an urology residency program and 88% applicants (n = 33) with 83% interviewers (n = 6) were in favor of utilizing VI as a screening tool adjunct to FTFI in the future [15]. Pasadhika et al., concluded that VI was an acceptable alternative to FTFI because half of their applicants (n = 18) expressed benefit of scheduling more interviews with VI [16]. Melendez et.al (n = 10) and Liman et al. (n = 8), both concluded that VI could be used as a screening tool but not as a replacement for FTFI [17, 18].

## Virtual platforms

Skype [26] was the most commonly used virtual platform (n = 11) [3, 5, 7, 9–11, 13–17]. Of these 11 studies using Skype as their virtual platform, 2 studies reported giving applicants an

option between Skype and FaceTime [27]. [9, 11] Remainder studies used different virtual platforms such as Zoom [2, 28] and vidyo [4, 29]. See Fig 4.

## Technology challenges

Some studies shared encountering technical issues during VI. Three studies reported technical issues from the interviewing institutions [2, 11, 14]. Vining et al., used Zoom and reported connectivity problem for 1 out of 12 interviewers [2]. Temple et. al, offered both Skype and FaceTime to their applicants to accommodate their preferences. They experienced difficulty with FaceTime due to challenges in internet connection specifically from their basement offices. During two of their Skype interviewing sessions, they encountered connection problems requiring follow up telephone call for interview completion [11]. Edje et al., used Skype and experienced voice delay during their interviews [14]. There were four studies that reported technical issues by the applicants [9, 10, 12, 15]. Vadi et. al, reported that of the total applicants (n = 42) who interviewed using Skype or FaceTime, 3 (9.4%) reported difficulty in maintaining eye contact; 2 (6.3%) reported sub-optimal video quality; and 1 (3.1%) reported sub-optimal audio quality during their interview [9]. In the Williams study using Skype, of the total 8 applicants, one applicant reported poor video quality [10]. Daram et al. using Facetime, shared that a total of three applicants mentioned that their overall experience with technology was suboptimal; one applicant reported poor audio and five reported suboptimal audio; one applicant reported suboptimal video experience and one applicant reported suboptimal eye contact during interviewing [12]. Shah et al. experienced that some of their applicants were in-between other interviews and connected to Skype from hotels leading to a poorer connection quality with video interruption and call termination [15].

## Virtual opportunity for a program tour and trainee interactions

There were 4 studies that reported giving virtual institutional tours [7, 9, 10, 15]. Other studies relied on their institution web pages or paper information to provide program information. There were 2 studies that additionally presented the applicants with the opportunity to interact with program trainees virtually and with fellow applicants on the interview day [9, 15].

## Applicant opinions about VI

Most of the studies reported that their applicants were able to appropriately convey themselves through VI [2–4, 7, 10] (Fig 4). Vining et al. (n = 16) reported that majority of their applicants (81%) were able to convey themselves well, were comfortable with VI (93%) and reported a good grasp of the program they were applying for (100%); vast financial benefits and time savings with reduced travel-related stress were plus points of VI with lack of adequate exposure to the location and in-person interaction being the top most disadvantages [2]. Chandler et al. (n = 20), published that most of their applicants shared that they could represent themselves well to the interviewers during their VI and were able to see if the program was a good fit for them; the common drawbacks being interaction with limited faculty, inability to tour the hospital, lack of opportunity to observe trainee-faculty interaction thus collectively limiting applicants' ability to assess the program's fit [3]. In Miotto et al. (n = 8) study, 12.5% applicants strongly agreed and 50% somewhat agreed that VI gave them an opportunity to adequately understand the program and about 87% applicants agreed that VI helped them decide about the program's fit [4]. Healy et al. (n = 47) reported that 85% applicants believed that VI allowed them to present themselves satisfactorily to the institution and an equal amount shared that VI gave them a good understanding of the program; 81% applicants were reassuringly able to rank the program solely based on their VI experience [7]. Vadi et al. (n = 42), found that the

commonest reasons their applicants chose VI were conflict of interview dates with different programs (31%), location limitations with travel concerns (28%), and financial constraints (25%); the commonest reasons for their applicants who chose FTFI, (n = 71), were an inclination to interact with trainees (45%), geographic proximity (30%) and a preference for in-person institutional tour (11%) [9]. Williams et al. recorded that 5 out of their total 6 applicants commented that VI adequately helped them rank the program and all of the 6 applicants shared that they would rank a program in which they had participated exclusively through VI [10]. Edje et al., (n = 9) reported that the commonest reasons for their applicants to choose VI included logistical ease, cost, innovative appeal and convenience [14]. See Fig 4.

## Institutional/Program opinions about VI

Vining et al., found that their interviewers, (n = 12), welcomed the idea of being able to interview applicants without necessarily having to be on-site [2]. All of the interviewers from the Chandler et al. study, (n = 3), shared that they were able to represent the program adequately to their applicants to help them decide if the program was the right fit. All of their interviewers agreed that technology worked well, and that VI was effective in helping them rank the applicants appropriately. They also reported that although VI could not possibly replace FTFI, it could act as a good screening tool [3]. Academic surgeons who acted as interviewers in the Healy et al. study, (n = 8), reported that they preferentially welcomed the reduced time that they eventually allocated to interview days as a result of VI compared to FTFI [7]. Interviewers from the Williams study, (n = 4), expressed their concern that their applicants who attended the VI were unable to tour the campus on their interview day [10]. Daram et al. conducted both VI and FTFI simultaneously with one interviewer at an off-site location while the other five faculty members conducting FTFI [12]. Shah et al. concluded that the flexibility offered by VI was helpful for both, applicants and faculty, to schedule interviews at a convenient time and also to cut down on the interview costs [15]. Melendez et al.'s interviewer (n = 1) reported that VI was beneficial to schedule interviews with applicants in their down time (while on vacation or at home at their convenience) and he also observed that the applicants appeared more relaxed during VI [17]. Edje et al. reported that VI was economically efficient, and their applicants reported cost savings ranging from $200 to $700.00 as a result of VI [14]. Pasadhika et al., also reported cost savings for their applicants and further added that VI was economically efficient only if it did not have an added in-person program tour clubbed together [16]. Their program also did not find any statistically significant difference in ranking the top applicants by VI as compared to FTFI.

## Discussion

We conducted this systematic review of the peer-reviewed literature to examine the use of synchronous VI in the interviewing process. This is the first systematic review researching available reports in literature about VI. Deriving robust conclusions from the studies included in this systematic review is challenging because of their limited number, their small sample size, multiple variables in their study design and the lack of comprehensive inclusion of many of the typical components of an interview day. The results suggest that VI has been explored by a handful of programs in the past as an adjunct or an alternative to FTFI [2–18] with most participating applicants and interviewing programs expressing reservations about VI's use as an alternative to FTFI and identifying VI to be only effective as a screening tool to shortlist applicants for FTFI [2–4, 11, 14, 15, 17, 18]. Applicants and faculty from only 2 studies thought that VI could be used independently [12, 16] providing hope for this unconventional method of interviewing. See Fig 4. We found that most applicants and interviewing faculty described lack

of opportunity of exploring the campus, location and the city, inability to adequately interact with trainees and faculty members of the institution, the accompanying anxiety of applicants as they doubt their ability to fully express themselves in VI as compared to FTFI, and the lack of clarity about how institutions finally rank the applicants opting for VI as compared to FTFI, as the major disadvantages of VI.

Supported by our findings of this systematic review and the fact that existing COVID-19 pandemic has necessitated all institutions to resort to VI this interviewing season, we propose supportive strategies to help prepare applicants and institutions for the successful implementation of VI.

## Strategies for institutions

Institutions should update their web pages to best reflect their current status to help applicants get as much information as possible offline. The web page should include frequently answered questions to guide applicants. Institutions should record high quality, comprehensive virtual tours of the interviewing facility taking into consideration the helplessness of existing applicants to tour the facility in-person and share with the applicants. Additionally, tour of the neighborhood and the city should also be available for the applicants to get a good feel of surrounding geography. If possible, videos on nearby daycare and children's activities should also be included to cater to applicants with children. Sharing brief recorded video introductions of key staff members with applicants will be well received. Programs should schedule a "test run" before the actual interview day to allay the anxiety and check for any technical error on either side. Institutions should use a virtual platform that will allow free account setup, easy access ability and an opportunity to create breakout rooms. Breakout rooms will allow organizers to help applicants easily transition into their multiple interviews and help facilitate interactions with potential future colleagues and fellow applicants, a key advantage of FTFI. Adequate time should be kept in between interviews to allow applicants to seamlessly express themselves during VI. Additional technical support should be kept stand-by on the interview day for troubleshooting any unanticipated glitches. Institutions should make every effort to be uniform with all applicants in the interviewing and ranking process to promote fairness. Interviewers should be understanding of any unanticipated technological problem during the interview, and distractors such as technical problems, lighting, background of the applicants etc. should not affect the final ranking process. Institutions should allow adequate time to assess the communication and interpersonal skills and "screen presence" in VI. They should offer applicants flexibility with scheduling interviews during downtime (evenings, weekends, holidays etc.) and should be thoughtful about applicant's personal schedule prior to the interview. For example, Chandler et al. noted that applicants may appear more fatigued or stressed during VI if their interview was scheduled after their work hours compared to FTFI where the applicant is presumably free of their home institution's responsibilities [3].

## Strategies for applicants

Applicants should thoroughly review the institution's web page and any other information source (virtual tour, neighborhood tour, faculty and staff introductions,) to get a better understanding of the institution. Applicants should perform personal research to find missing details. Applicants should not hesitate to reach out to institutions with questions that would help them assess the institution's fit. Applicants should request for a "test run" prior to the final interview to assess for potential technical errors. On the day of the interview, applicants should be prepared with good internet connection, laptop and a high-quality webcam for better results. Applicants should try and create accounts on all available virtual platforms to offer

flexibility to hosting institutions. They should find a "professional" looking space in their home in preparation for VI. Lighting should preferably be over the head and not behind, to avoid screen glare. Applicants should keep their background subtle and consider having neutral background colors with softer paintings. Applicants should arrange for a quiet environment during the VI. They should dress professionally as they would for FTFI. Camera should be positioned at eye level. Applicants should avoid looking at the screen and instead look into the camera as if they were speaking and making eye contact with a FTFI interviewer. They should avoid distracting body movements. For example, one study reported that an applicant who positioned the camera such that it made the applicant appear looking downward and who also swiveled nervously in a wheeled chair had a less successful interview [10]. Applicants should promptly report technical problems to the VI organizer and seek help.

## Limitations

Our study is not without limitations. Despite our diligent effort to include all relevant search terms, we may have accidentally excluded keywords and thus relevant studies. Although we kept our inclusion criteria broad, we did not include conference abstracts limiting our sample size. There are very few studies demonstrating an impact of VI with complete replacement of FTFI and therefore may fail to reflect the true potential of VI in the interviewing process. The studies included had a small sample size making it challenging to accurately interpret findings.

## Conclusions

This is the first systematic review demonstrating experiences from applicants and institutions on VI. Despite described limitations, our systematic review adds to the important role of VI in the interviewing process as institutions transition to VI setting due to travel restrictions imposed by COVID-19. This research has the potential of helping applicants and institutions in preparing themselves for upcoming VI process. Strategies to help prepare applicants and institutions in conducting VI described by the authors may help formulate best practices for the upcoming interviewing process.

## Implications for future research

There is dearth of evidence in literature about the efficacy of VI. VI holds an immense potential in interviewing. COVID-19 travel restriction has now forced institutions to conduct VI this interviewing season. This presents an opportunity for potential research at multi-institutional level to gain better understanding of the efficacy of VI and demonstrating the true impact of VI as it completely replaces FTFI in the interviewing process in all fields related to healthcare across the world. Research evaluating the role of VI during COVID-19 will direct us in appropriate inclusion of VI in the interviewing process in the absence of travel limitations.

## Supporting information

**S1 Appendix. MeSH used for PubMed database.**
(DOCX)

**S2 Appendix. PRISMA 2009 checklist.**
(DOC)

## Author Contributions

**Conceptualization:** Sonal Chandratre, Aamod Soman.

**Data curation:** Sonal Chandratre, Aamod Soman.

**Formal analysis:** Sonal Chandratre, Aamod Soman.

**Investigation:** Sonal Chandratre, Aamod Soman.

**Methodology:** Sonal Chandratre, Aamod Soman.

**Project administration:** Sonal Chandratre.

**Resources:** Sonal Chandratre.

**Software:** Sonal Chandratre.

**Supervision:** Sonal Chandratre, Aamod Soman.

**Validation:** Sonal Chandratre.

**Visualization:** Sonal Chandratre.

**Writing – original draft:** Sonal Chandratre.

**Writing – review & editing:** Sonal Chandratre, Aamod Soman.

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
