## [Decision Letter · Decision Letter 0]

23 Nov 2020

Preparing For The Interviewing Process During Coronavirus Disease-19 Pandemic: Virtual Interviewing Experiences Of Applicants And Interviewers, A Systematic Review

PONE-D-20-34511

Dear Dr. Chandratre,

We’re pleased to inform you that your manuscript has been judged scientifically suitable for publication and will be formally accepted for publication once it meets all outstanding technical requirements.

Kind regards,

Andrej M Kielbassa, Prof. Dr. med. dent. Dr. h. c.

Academic Editor

PLOS ONE

Additional Editor Comments (optional):

This submitted draft would seem ready to proceed, congratulations. Please double check the proofs for minor shortcomings.

Warm regards and stay healthy, please.

Reviewers' comments:

Reviewer's Responses to Questions

**Comments to the Author**

1. Is the manuscript technically sound, and do the data support the conclusions?

Reviewer #1: Yes

2. Has the statistical analysis been performed appropriately and rigorously? 

Reviewer #1: N/A

3. Have the authors made all data underlying the findings in their manuscript fully available?

Reviewer #1: Yes

4. Is the manuscript presented in an intelligible fashion and written in standard English?

Reviewer #1: Yes

5. Review Comments to the Author

Reviewer #1: This is considered a sound piece of work, easily intelligible, and worth following. With your Abstract section, please refer to the PRISMA statement when reporting your study. Due to timely topic, recommendations is "accept", and proceeding should be without delay.

6. PLOS authors have the option to publish the peer review history of their article (what does this mean?). If published, this will include your full peer review and any attached files.

Reviewer #1: No

---

## [Editor Report · Acceptance letter]

25 Nov 2020

PONE-D-20-34511 

Preparing For The Interviewing Process During Coronavirus Disease-19 Pandemic: Virtual Interviewing Experiences Of Applicants And Interviewers, A Systematic Review 

Dear Dr. Chandratre:

I'm pleased to inform you that your manuscript has been deemed suitable for publication in PLOS ONE. Congratulations! Your manuscript is now with our production department. 

Kind regards, 

on behalf of

Prof. Dr. med. dent. Dr. h. c. Andrej M Kielbassa 

Academic Editor

PLOS ONE